# Effects of Community-Led Total Sanitation (CLTS) Boosting and Household Factors on Latrine Ownership in Siaya County, Kenya

**DOI:** 10.3390/ijerph20186781

**Published:** 2023-09-18

**Authors:** Job Wasonga, Kazuchiyo Miyamichi, Mami Hitachi, Rie Ozaki, Mohamed Karama, Kenji Hirayama, Satoshi Kaneko

**Affiliations:** 1Graduate School of Biomedical Sciences, Nagasaki University, Nagasaki 852-8523, Japan; jwasonga@hotmail.com; 2Department of Ecoepidemiology, Institute of Tropical Medicine, Nagasaki University, Nagasaki 852-8523, Japan; kk_miyamichi@nagasaki-u.ac.jp (K.M.); mhitachi@nagasaki-u.ac.jp (M.H.); rieozaki1979@gmail.com (R.O.); 3Centre for Public Health Research, Kenya Medical Research Institute (KEMRI), Nairobi 20752-00202, Kenya; mhmdkarama20@gmail.com; 4AMREF Health Africa Ethics and Scientific Research Committee, Amref Health Africa, Nairobi 27691-00506, Kenya; 5School of Tropical Medicine and Global Health, Nagasaki University, Nagasaki 852-8523, Japan; hiraken@nagasaki-u.ac.jp

**Keywords:** community-led total sanitation, open defecation, latrine possession, Kenya

## Abstract

Community-led total sanitation (CLTS) is a widely used approach for enhancing sanitation practices. However, the impact of boosted CLTS on household latrine ownership has not been adequately evaluated. This study aims to investigate the factors associated with latrine possession among households, with a specific focus on single and CLTS-boosting implementation. A community-based repeated cross-sectional study was conducted in Siaya County, Kenya, involving 512 households at the baseline and 423 households at the follow-up. Data were analyzed using the mixed-effects logistic regression model. At the baseline, latrine possession was significantly associated with CLTS implementation (adjusted OR [aOR]: 3.01; 95% confidence interval [CI]: 1.41–6.44), literacy among households (aOR: 1.83; 95% CI: 1.12–2.98) and higher socioeconomic status (SES) (second level: aOR: 2.48; 95% CI:1.41–4.36, third level: aOR: 3.11; 95% CI: 1.76–5.50, fourth level: aOR: 10.20; 95% CI: 5.07–20.54). At follow-up, CLTS boosting (aOR: 7.92; 95% CI: 1.77–35.45) and a higher SES were associated with increased latrine ownership (second level: aOR: 2.04; 95% CI: 0.97–4.26, third level: aOR: 7.73; 95% CI: 2.98–20.03, fourth level: aOR: 9.93; 95% CI: 3.14–28.35). These findings highlight the significant role played by both single and CLST boosting in promoting universal latrine ownership and empowering vulnerable households to understand the importance of sanitation and open defecation-free practices.

## 1. Introduction

Sanitation is a crucial aspect of human health and dignity. Open defecation (OD) poses risks to human health, including diarrheal diseases and childhood stunting [1,2,3]. Additionally, poor sanitation has adverse effects on social and economic development, as well as gender equity [4,5,6,7]. Despite global efforts, two billion people lack access to essential sanitation services, and 673 million individuals still practice open defecation [8]. The absence of primary and acceptable sanitation constitutes a significant public health and social problem in many low- and middle-income countries (LMICs), underscoring the urgent need for sanitation improvements to save lives and foster individual and social development. 

Numerous approaches have been implemented globally to address the challenges of poor sanitation. Community-led total sanitation (CLTS) has emerged as a popular approach in rural areas of low-income countries. CLTS is a social motivation approach that helps communities address open defecation through promoting sanitation that emphasizes community participation. Through community empowerment, CLTS employs participatory tools to ‘trigger’ emotions of shame, disgust, fear, and self-respect, convincing the community to halt open defecation and construct and maintain their latrines. The successes are achieved by engaging communities in ‘shaming and coercion’ to make them build latrines without external subsidies [9,10,11,12]. CLTS encourages communities to construct latrines using locally available resources and materials and commit to not practicing open defecation (OD) [13]. The CLTS approach, initially developed by Kamal Kar, aims to create collective behavioral change through a simple, facilitated process of creating disgust with open defecation. This is achieved through community self-regulation and enforcement, where the responsibility for progress is primarily left to the community [14]. 

The CLTS process involves facilitators catalyzing community discussions, highlighting the health risks of unsanitary practices, and making them aware of their open defecation sites through a transect walk, leading to spontaneous latrine construction. By empowering communities to take ownership, CLTS promotes sustainable behavior changes [15,16]. Unlike previous infrastructure-centered or educational interventions, CLTS focuses on motivating and encouraging participants to bring about behavioral change and construct latrines using locally available resources [13,17]. Over 60 countries have adopted CLTS to enhance sanitation access and achieve an open defecation-free (ODF) status [18].

Despite the widespread use of CLTS and the evidence that CLTS improves health and sanitation [19,20,21,22,23,24], concerns and limitations have been identified. First, there is a lack of rigorous evidence regarding the impact of CLTS on latrine ownership. Previous studies have explored various determinants of household latrine ownership, including socio-demographic, environmental, and individual psychological factors [16,25,26,27,28]. Although some studies suggest a positive correlation between CLTS implementation and latrine coverage, existing evidence primarily relies on gray sources in the literature, resulting in limited and insufficiently supported information for policymakers [29].

The second issue pertains to the effectiveness of CLTS interventions in promoting household latrine ownership. While some communities have shown success, the overall progress toward universal latrine ownership remains slow [29,30]. In rural areas of Kenya where this study was undertaken, many communities still face sanitation challenges. It has been reported that 15% of households in 2015 and 16% in 2022 use improved facilities while others use shared latrines (with other households), begging the question of what the remaining households use. In addition, pit latrines, suspended latrines, and bucket latrines without a slab or platform were reported to be used by 39% of households in 2015 and 40% in 2022 [31]. This shows that a minimal significant improvement has occurred since 2015 regarding sanitation and latrine construction and improvement [32]. Although open defecation decreased from 13% in 2015 to 9% in 2022, more than eight million Kenyans still practice open defecation, spreading diarrheal diseases and incurring significant economic losses [33]. The Kenyan government adopted CLTS in 2011 to scale up latrine coverage and eradicate open defecation by 2013 [34]. However, the Kenyan government’s efforts have not achieved much, and in some areas, open defecation is still rampant despite CLTS having been implemented. 

Some studies have shown the need for interventions to address the collapse and difficulty of rebuilding latrines after CLTS to achieve ODF and also how, in areas with low latrine coverage and high OD rates, sanitation interventions that emphasize CLTS and improved latrines can have significant benefits in achieving ODF [35,36]. Due to a similar situation in Kenya, some regions performed a second round of CLTS interventions, known as CLTS boosting. CLTS boosting was introduced to enhance the impact of CLTS and ensure ODF villages.

This study aims to assess the impact of CLTS and its boosting on latrine ownership, taking into account factors associated with ownership. By addressing these knowledge gaps, this research aims to contribute to improving sanitation practices and inform policy decisions in the context of CLTS interventions.

## 2. Materials and Methods

### 2.1. Study Setting and Participants

A community-based repeated cross-sectional study was conducted in Siaya County in western Kenya. In Kenya, there are 47 counties divided into sub-counties, wards, and villages. Villages are the smallest units and are at the lowest administrative level. Siaya County is administratively divided into six sub-counties (Figure 1). Out of the six sub-counties, two sub-counties (Alego Usonga and Rarieda) were randomly selected from among four sub-counties where the county government had initiated CLTS, and one sub-county (Bondo) was selected from among two sub-counties where CLTS had not been initiated. These sub-counties were selected, taking into account the travel distances involved when conducting the survey. In addition, three villages were randomly selected from each of the three selected sub-counties, resulting in nine villages as study sites (Figure 2). Since CLTS was implemented at the village level, one village was selected in the sub-county where CLTS was initiated but not implemented (Kametho A village in Rarieda sub-county).

The study area is predominantly inhabited by the Luo tribe with a polygamous culture. The Luo are typically patrilineal and virilocal (living in man’s family place). They live in family homesteads, “*dala*”, which traditionally comprise a male head of the homestead, his wives or wife and their children, and his married son’s families forming several households within the homestead or compound. When the husband has several wives, each wife has her own house within the homestead, and the first wife’s house is prominent and becomes the reference point of the homestead. In this study, we treated each wife’s house as a household and selected the first wife’s household for interviews and observations. This is because of the significant status and role of the first wife in this polygamous system, commonly known as the “senior wife” or “head wife” [37].

### 2.2. Data Collection

Data were collected using structured questionnaires (Appendix A) between May and July 2016 as a baseline survey and March and April 2018 as a follow-up survey, respectively (Figure 2). The questionnaire was administered by trained local data collectors accompanied by local community health volunteers to assess the household status (demographic and socioeconomic variables) and latrine possession by the household. The households surveyed were the first wife’s household of a homestead, and the first wife was selected as the respondent. Demographic variables included the household head or representative’s marital status and reading ability, the household population, and the presence of children under five. Regarding socioeconomic status (SES), we asked about the utilization or possession of the following items: a mobile/telephone, TV, sewing machine, posho mill, ox-plough, bicycle, radio, motorcycle, motor vehicle, type of floor, as well as roof, wall, and cooking fuel. The possession or ownership of poultry, cattle, goats, sheep, pigs, and donkeys was also asked during the interview.

The status of CLTS implementation was identified for each village because CLTS was conducted at the village level according to the county government’s action plan. In the baseline survey, CLTS implementation was defined as CLTS implemented and CLTS not implemented. At the time of the follow-up survey, all villages had received at least one CLTS intervention, and some had received a CLTS boost. Therefore, to assess the effect of the CLTS boost on latrine ownership, we identified this by registering whether villages had received a boost or a second CLTS intervention since the baseline survey.

### 2.3. Data Analysis

A mixed-effects logistic regression model was employed to evaluate the factors associated with latrine possession, with household latrine possession as the outcome variable (possess or not). Furthermore, to add an explanatory variable, a new category was created performing the principal components analysis (PCA) on household asset information, which was then incorporated into the analytical model to measure socioeconomic status (SES). In the PCA, household assets were classified as owned or not owned. Livestock values were converted according to the monetary value of each animal, which was then summed to obtain the value of livestock ownership for each household. The monetary value of each livestock was converted as follows: KSH 500 (Kenyan Shilling) per poultry, KSH 4000 per goat, KSH 3000 per sheep, KSH 15,000 per cow, KSH 3000 per pig, and KSH 20,000 per donkey. KSH 100 is equivalent to USD 0.7. These monetary values were based on the selling price in the local area (markets) at the time of conducting this study. The households were then divided into three groups according to their distribution, which fed into the PCA results. 

The score derived from the first PCA component was taken as the wealth index of each household’s SES [38]; households were then divided into four groups based on their wealth index, with the first quantile SES being the poorest group and the fourth quantile of SES the wealthiest group. In terms of household size, households were divided into two groups (≤4 or >4) based on the average household size of Siaya County [39].

The data were compiled based on latrine ownership and household information from the households surveyed using the baseline and follow-up surveys. A Chi-square test was conducted to assess the potential bias between latrine ownership and the corresponding household information.

A mixed-effects logistic regression analysis was performed using univariate and multivariate models to calculate crude and adjusted odds ratios for the variables included in the model. The best model for multivariate analysis was selected based on the lowest Akaike information criteria (AIC). The full model included variables such as reading ability, the wealth index of SES, and the status of CLTS implementation (CLTS implementation for the baseline survey and CLTS boosting for the follow-up survey) as fixed effects, while the village variable was treated as a random effect. All statistical analyses, including principal component analysis (PCA) to extract the first principal component for SES classification using the ‘pca’ command, mixed-effects logistic regression analysis using the ‘xtmelogit’ command, the number of adaptive Gaussian quadrature integration points set to seven to report crude odds ratios (cOR) and adjusted odds ratios (aOR) with a 95% confidence interval (CI), were performed using Stata 14 (Stata Corp LLC, College Station, TX, USA). *p*-values less than 0.005 were considered statistically significant. For the creation of the study site map, QGIS (3.28, Windows 10) (Open Source Geospatial Foundation, available online: qgis.org) was utilized, and community boundaries were delineated using the GADM database (available online: gadm.org (accessed on 15 June 2023)).

### 2.4. Ethical Consideration

The study was approved by the Scientific and Ethics Review Unit of Kenya Medical Research Institute (SCC No. CPHR/006/3174 approved on 28 January 2016). Surveys, interviews, and observations were conducted only after obtaining informed consent from the research participants, and strict anonymity was maintained. The data collectors read out a written consent form to the participants, who gave consent for each interview. In addition, the consent form was given to the participants/households for their reference. Written consent was obtained at the start of the baseline survey, and verbal consent was obtained at the follow-up survey to confirm willingness to participate again (Appendix A).

## 3. Results

### 3.1. Participants’ Characteristics

During the baseline study, all 514 first-wife households from the nine villages in Siaya County participated. For the analysis, 512 households in the baseline survey and 423 in the follow-up survey met eligibility criteria (Figure 3). In 2016, during the baseline survey, four villages had not yet implemented CLTS, while five villages had implemented CLTS. During the follow-up survey in 2018, all villages had received at least one CLTS intervention, and only three implemented a second CLTS, known as CLTS boosting (Figure 2).

Table 1 shows the characteristics of households based on latrine ownership status. In the baseline survey, over two-thirds (69.5%) of the 512 households owned latrines. Concerning other characteristics, households with marital status, reading ability, SES, and the sub-county (with or without CLTS implementation) were significantly associated with latrine ownership. At follow-up, 86.8% of households owned latrines, and SES and sub-county (associated with CLTS boosting status) were significantly associated with latrine ownership.

### 3.2. Factors Associated with Latrine Possession at the Baseline

According to the best multivariate mixed-effects logistic regression model, the baseline survey showed that CLTS implementation status, reading ability, and SES were significantly associated with household latrine possession (Table 2). Households with CLTS were 3.01 more likely to own a latrine than those without CLTS (95% confidence interval [CI], 1.41–6.44), and literate households were 1.83 times more likely to own a latrine than non-literate households (95% CI: 1.12–2.98). Regarding SES, the adjusted odds ratio for latrine ownership significantly increased with higher SES levels (second level: aOR: 2.48; 95% CI: 1.41–4.36, third level: aOR: 3.11; 95% CI: 1.76–5.50, fourth level: aOR: 10.20; 95% CI: 5.07–20.54). Marital status was significantly associated with latrine possession in the univariate model (crude odds ratio [cOR]: 1.94; 95% CI: 1.27–2.95) but not in the best model.

### 3.3. Factors Associated with Latrine Possession at Follow-Up

According to the best multivariate mixed-effects logistic regression model using follow-up data (Table 3), households with CLTS boosting were 7.92 times more likely to own a latrine compared to those with CLTS non-boosting (95% CI: 1.77–35.45). Furthermore, although there were no statistically significant differences between the poorest and second poorest households, a similar trend to the baseline survey was observed: higher SES tended to be associated with higher latrine possession (second level: aOR: 2.04; 95% CI: 0.97–4.26, third level: aOR: 7.73; 95% CI: 2.98–20.03, fourth level: aOR: 9.93; 95% CI: 3.14–28.35). Additionally, literate households’ likelihood of owning latrines was not statistically significant (aOR: 1.18; 95% CI: 0.60–2.32).

## 4. Discussion

This study aimed to investigate the associated factors of latrine possession, including single and boosted CLTS implementation. The results of this study revealed that households’ latrine ownership was significantly related to CLTS implementation and boosting, households’ reading ability, and SES. Specifically, households with CLTS boosting presented a higher odds ratio for latrine possession compared to single CLTS implementation (Table 2 and Table 3). This study confirms the positive association between CLTS implementation and latrine possession, as reported by previous studies [29]. Furthermore, our results suggest that not only single CLTS implementation but also CLTS boosting is essential to achieve the ultimate goal of CLTS in catalyzing universal latrine ownership and ODF status. 

Table 3 displays the survey and analytical findings following the second CLTS implementation in relation to the CLTS boosting effect. Villages that had CLTS implemented for the first time in 2018 and those that had CLTS implemented for the second time (boost) were included in the analyzed data. The odds ratio of latrine ownership decreased in villages where CLTS was first implemented in 2015 and 2018 compared to villages where CLTS was first implemented in 2014, according to the crude OR, with the year of first CLTS implementation as the variable. The lower OR was because all villages where the first CLTS was conducted in 2014 had a second CLTS implemented in 2018, meaning those communities improved their latrines. This is due to the fact that when comparing villages where a second CLTS was implemented (with the first year of implementation being 2014) to villages where the first CLTS was implemented in 2015 and 2018, a similar effect was discovered on the ownership of similar latrines. This was evident from the results of the analysis from the year of the first CLTS in the crude OR. Additionally, the initial CLTS implementation year variable (included in the CLTS twice) was removed from multivariate analysis via variable selection. This outcome demonstrates and supports the importance of encouraging and motivating the community to construct latrines and guarantee that villages are free of open defecation. The results of this study concur with those of a study conducted in Ethiopia, which found a substantial correlation between supervision following intervention and latrine use [40]. 

In addition to the results obtained in this study, the lower latrine coverage among lower SES households has been shown in previous studies [30,40]. The low latrine coverage among people with low incomes could be explained in terms of the cost of latrine construction. Jenkins and Beth showed that the cost related to household latrines could be a significant barrier to latrine construction and ownership [16]. More so, the cost of constructing a public latrine in a community can be funded by external or co-financing, whereas self-financing is generally required for the individual latrines of households based on CLTS policy [13]. Thus, whereas several studies have indicated that individual beliefs about cost and benefit are predictors of latrine ownership [25,30], the issue of motivation plays an important role. It can be assumed that there are differences in motivation between households depending on their SES, in as much as CLTS tries to motivate the community. Additionally, it may be challenging for lower SES families to construct their own latrines with a limited household budget, regardless of their desire. Hence, specific support is needed to facilitate latrine construction for households in lower SES.

Furthermore, as has been shown in other studies, reading ability was identified as an associated factor for latrine possession in this study [28,30]. The relationship between household latrine possession and knowledge about hygiene and sanitation has been well documented [25,30,41]. Thus, illiterate people may not be able to fully benefit from advertising and educational interventions using written materials. However, the association between reading ability and latrine ownership was not statistically significant in the follow-up survey. The effects of social networks could explain this. Shakya et al. demonstrated that households are more likely to have their own latrine if their social networks own one [41]. In addition, this relationship is stronger among households with similar characteristics, such as educational level. In this study, latrine coverage at the follow-up survey was higher than that at the baseline. Consequently, relatively more illiterate households owned a latrine compared to the baseline. As a result, the effect of household characteristics, including reading ability, could become weaker as latrine coverage improves in the community.

Regrettably, this study could not observe each latrine owned by participants, although it was confirmed verbally. Thus, the coverage of latrine possession may be overestimated due to the shame of ODF. In addition, latrine quality, maintenance, and utilization were not assessed in this study. Garn et al. indicated that latrine ownership and coverage do not necessarily lead to an equal increase in latrine utilization [42]. Hence, further studies may be necessary to assess the factors influencing latrines’ utilization and quality. In particular, evaluating the impact of CLTS boosting on latrine utilization and quality is vital to informing policy decisions. Understanding the factors that affect the effective utilization and maintenance of latrines is crucial for sustainable sanitation interventions. Therefore, future research should focus on examining latrine utilization’s determinants and latrines’ quality. Specifically, investigating the role of CLTS boosting in improving latrine utilization and assessing the factors contributing to latrine maintenance and sustainability could provide valuable insights for designing effective sanitation programs and policies. Lastly, this study did not evaluate the change in collective behavior or social norms essential to enhance individual behavior change and sustainability [41,43,44]. Further studies are needed to explore the impact of boosted CLTS on collective behavior change or social norms.

## 5. Conclusions

The results of this study provide evidence that CLTS implementation, households’ reading ability, and SES are associated with latrine ownership. This study is the first to evaluate the relationship between CLTS boosting and latrine ownership. The findings suggest that CLST boosting is essential in promoting universal latrine ownership. Furthermore, there is a need to enhance the development of CLTS interventions to effectively reach and empower vulnerable households, enabling them to understand the significance of sanitation and ODF practices.

## Figures and Tables

**Figure 1 ijerph-20-06781-f001:**
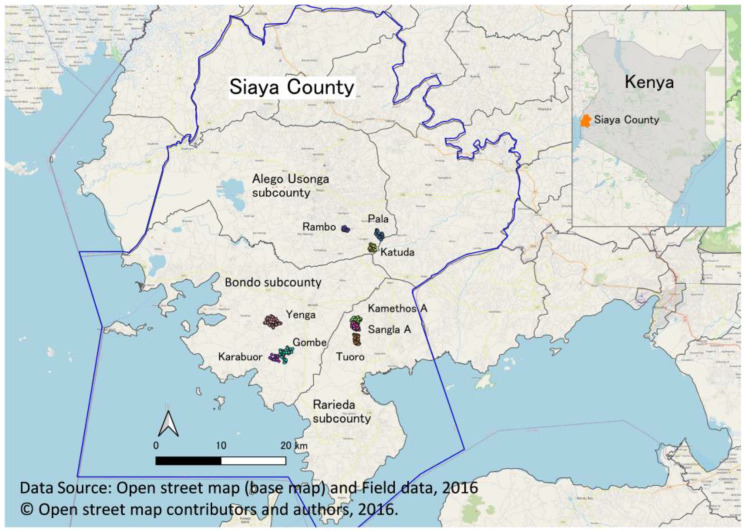
Study site and design of study villages. Three sub-counties were selected: Alego Usonga and Rarieda (where CLTS had already started) and Bondo (where CLTS had not started) in 2016. Three villages were selected from each of these three sub-counties (9 villages in total).

**Figure 2 ijerph-20-06781-f002:**
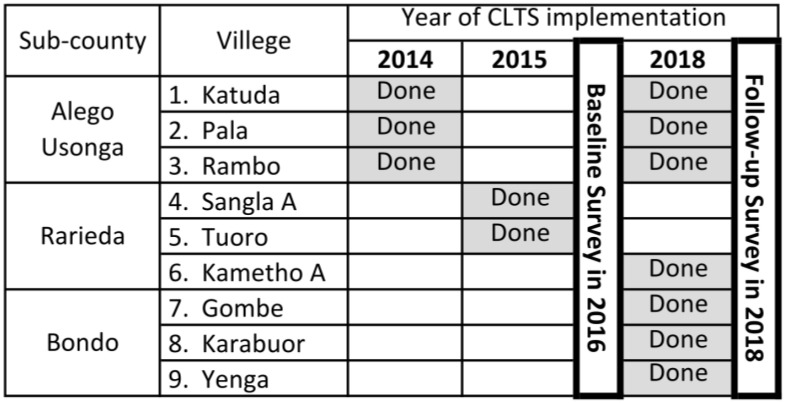
Timing of CLTS implementation in the nine study villages and the schedule of two surveys: Villages that received CLTS for the second time in 2018 were considered as boost CLTSs in this study; villages that received CLTS for the first time in 2018 were only considered as the first CLTS.

**Figure 3 ijerph-20-06781-f003:**
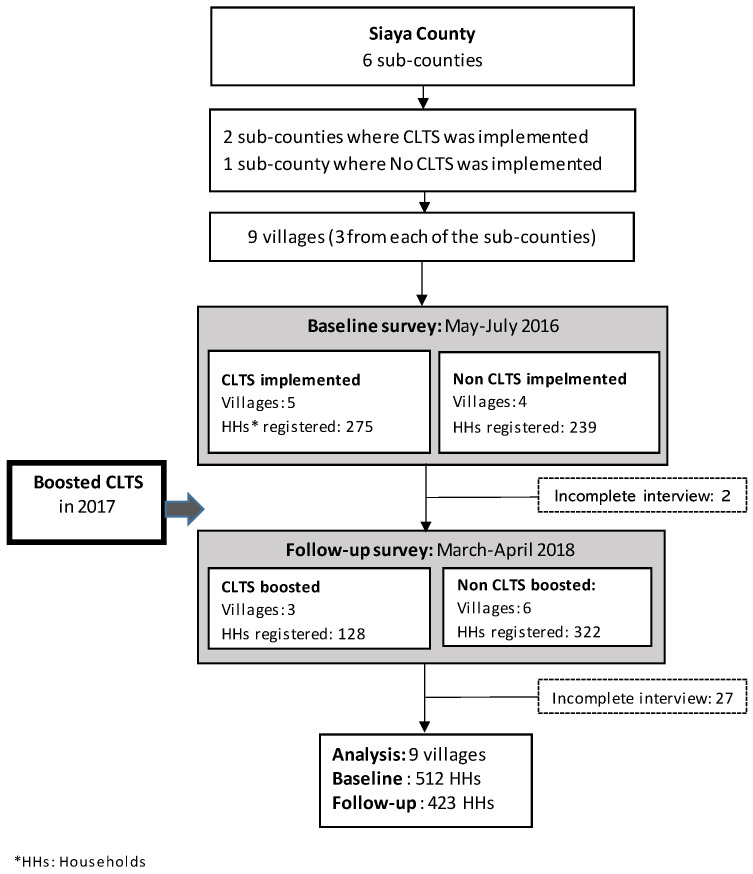
Flow diagram of enrolled participants and study procedure.

**Table 1 ijerph-20-06781-t001:** Distribution of latrine ownership by characteristics of households.

		Baseline Survey (N = 512)						Follow-Up Survey (N = 423)					
Characteristic		Toilet Possessed	(%)	No Toilet Possessed	(%)	Total	χ2	Toilet Possessed	(%)	No Toilet Possessed	(%)	Total	χ2
							*p*-Value						*p*-Value
CLTS status at baseline							<0.001						
	Yes	142	59.9%	95	40.1%	237		-	-	-	-		
	No	214	77.8%	61	22.2%	275		-	-	-	-		
CLTS status at follow-up													<0.001
	Yes	-	-	-	-			251	82.8%	52	17.1%	303	
	No	-	-	-	-			116	96.7%	4	4.0%	120	
Marital status							0.003						0.11
	Not married	95	60.5%	62	39.5%	157		106	82.8%	22	17.2%	128	
	Married	261	73.5%	94	26.5%	355		261	88.5%	34	11.5%	295	
Reading ability							<0.001						0.11
	No	63	55.3%	51	44.7%	114		118	83.1%	24	16.9%	142	
	Yes	293	73.6%	105	26.4%	398		249	88.6%	32	11.4%	281	
Household size							0.073						0.39
	≦4	184	66.2%	94	33.8%	278		161	85.2%	28	14.8%	189	
	4	172	73.5%	62	26.5%	234		197	87.6%	28	12.4%	225	
	Missing	-	-	-	-			9	100.0%	0	0.0%	9	
Presence of children U5							0.22						0.66
	No	203	67.4%	98	32.6%	301		201	87.0%	30	13.0%	231	
	Yes	153	72.5%	58	27.5%	211		161	86.1%	26	13.9%	187	
	Missing							5	100.0%	0	0.0%	5	
Socio-economic status (SES)							<0.001						<0.001
	Poorest	65	49.6%	66	50.4%	131		79	73.8%	28	26.2%	107	
	Second	86	68.8%	39	31.2%	125		90	84.9%	16	15.1%	106	
	Third	92	71.9%	36	28.1%	128		102	93.6%	7	6.4%	109	
	Fourth	113	88.3%	15	11.7%	128		96	95.0%	5	5.0%	101	
Sub-county							<0.001						<0.001
	Alego Usonga	105	70.5%	44	29.5%	149		116	96.7%	4	3.3%	120	
	Rarienda	109	86.5%	17	13.5%	126		91	77.8%	26	22.2%	117	
	Bondo	142	59.9%	95	40.1%	237		160	86.0%	26	14.0%	186	
Total		356	69.5%	156	30.5%	512		367	86.8%	56	13.2%	423	

**Table 2 ijerph-20-06781-t002:** Results of analysis of mixed-effects logistic regression models on latrine ownership at the time of baseline survey.

Variables		cOR ^a^	95% CI ^b^	*p*-Value	aOR ^c^	95% CI ^b^	*p*-Value
CLTS implementation							
	No	Reference			Reference		
	Yes	2.14	(1.03–4.45)	0.042	3.01	(1.41–6.44)	0.004
Marital status							
	Not married	Reference					
	Married	1.94	(1.27–2.95)	0.002			
Reading ability							
	No	Reference			Reference		
	Yes	2.25	(1.43–3.55)	<0.001	1.83	(1.12–2.98)	0.015
Household size							
	≦4	Reference					
	>4	1.49	(0.99–2.26)	0.057			
Presence of U5							
	No	Reference					
	Yes	1.37	(0.91–2.07)	0.13			
Socioeconomic status (SES)							
	Poorest	Reference			Reference		
	Second	2.7	(1.55–4.72)	<0.001	2.48	(1.41–4.36)	0.002
	Third	3.26	(1.85–5.72)	<0.001	3.11	(1.76–5.50)	<0.001
	Fourth	10.9	(5.43–21.89)	<0.001	10.2	(5.07–20.54)	<0.001

^a^: cOR: crude odds ratio. ^b^: 95% confidence interval. ^c^: aOR: adjusted odds ratio. Note: The best model for predicting latrine possession was chosen using the lowest Akaike’s information criterion (AIC).

**Table 3 ijerph-20-06781-t003:** Results of the analysis of mixed-effects logistic regression models on latrine ownership at the follow-up survey.

Variables		cOR ^a^	(95% CI) ^b^	*p*-Value	aOR ^c^	(95% CI)	*p*-Value
CLTS boosting							
	No (Single CLTS)	Reference			Reference		
	Yes (Boosting CLTS)	5.96	(1.66–21.48)	0.006	7.92	(1.77–35.45)	0.007
Year of first CLTS implementation							
	2014	Reference					
	2015	0.12	(0.03–0.45)	0.002			
	2018	0.21	(0.06–0.76)	0.017			
Marital status							
	Not married	Reference					
	Married	1.7	(0.92–3.14)	0.090			
Reading ability							
	No	Reference			Reference		
	Yes	1.64	(0.87–3.11)	0.127	1.18	(0.60–2.32)	0.622
Household size							
	≦4	Reference					
	> 4	1.44	(0.80–2.59)	0.229			
	Missing	-					
Presence of U5							
	No	Reference					
	Yes	0.98	(0.54–1.78)	0.948			
	Missing	-					
Socioeconomic status (SES)							
	Poorest	Reference			Reference		
	Second	2.03	(0.97–4.24)	0.060	2.04	(0.97–4.26)	0.059
	Third	7.88	(3.02–20.58)	0.000	7.73	(2.98–20.03)	0.000
	Fourth	10.3	(3.45–30.72)	0.000	9.43	(3.14–28.35)	0.000

^a^: cOR: crude odds ratio. ^b^: 95% confidence interval. ^c^: aOR: adjusted odds ratio. Note: The best model for predicting latrine possession was chosen using the lowest Akaike’s information criterion (AIC).

## Data Availability

Data are available upon reasonable request.

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
