# Peer review of "Effects of Community-Led Total Sanitation (CLTS) Boosting and Household Factors on Latrine Ownership in Siaya County, Kenya"

_ijerph, 2023, doi:10.3390/ijerph20186781_

Round 1
Reviewer 1 Report
·
Thanks to the authors for this manuscript. Please find below some comments to improve the manuscript
The authors note that ‘the national improved latrine coverage in Kenya is estimated at only 30% in 2015’. Could the authors present a more recent estimate? There are similar reports to the referenced reports that were released after 2015
· It is reported as ‘in Siaya County 73 in western Kenya, which is administratively divided…’. I suggest rephrasing this sentence to two sentences, i.e in Siaya County 73 in western Kenya. Siaya County is administratively divided….’
· Similarly suggest rephrasing line 74-77 to ‘Out of the six sub-counties, two sub-counties (Alego Usonga and Rarieda) were selected from among four sub-counties where the county government had initiated CLTS, and one sub-county (Bondo) was selected from among two sub-counties where CLTS had not been initiated. These sub counties were selected by taking into account the travel distances involved in conducting the survey’
· Line 79-80, three villages were randomly selected FROM…
· Overall, under the study setting, it would be useful for the authors to describe the administrative breakdown, to make it easier for the reader to understand. For example, let the readers know that Counties are divided into sub counties, and villages are the lowest administrative level. This makes it easier for readers to understand the different stages of sampling.
· In addition, please describe the cultural background of the community, for example, before stating that a household of the first wife was interviewed, it would be prudent to explain the cultural setting
· Line 81-82: What do the authors mean by a household of the first wife? Later on, the authors say ‘a compound homestead consists of several households’. Please define a household in your study
· I notice that the document has lots of spelling mistakes, grammatical errors or editorial mistakes. I recommend that the manuscript be checked for grammatical and editorial aspects
· Data collection: Please specify how the respondents were sampled/selected
· Line 106-107, it is not clear what this sentence means. Please clarify whether it means that in the follow up survey the outcome measure was CLTS boosting
· Line 119-122; how did you determine the value o livestock? Also, please revise this sentence to bring out the meaning better
· Please revise the term household population to household size
· Please describe the type of consent obtained from the respondents
· Please describe fully the descriptive analysis done, e.g. no mention of chi square test, yet table 1 shows Chi square results
Figure 2:
Please correct the grammatical mistakes, e.g Siayaya County
Please specify whether CLTS was implemented at Sub County level or at village level. Previously, it seemed as though it had been implemented at village level
What do ‘households registered’ mean? Was there registration of households, or is it in reference to households who were selected for the study?
· Line 181; the authors mention a bivariate model, which was not mentioned in the data analysis. Please clarify
· Table 2 and 3: Include the P-values
· Line 195: Please stick to the same words. Reading ability or literacy?
· Line 206: Check, association between CLTS implementation and what?
· This manuscript only investigates household factors that influence latrine ownership. The authors have clearly spelt out limitations of their study, including the fact that they did not observe the presence of the latrines. I think the title of the study should therefore be about household factors.
The English needs to be checked for errors
Author Response
We appreciate your valuable comments. We would like to respond to the reviewers' questions and comments and revise the manuscript.
Response to comments from Reviewer 1
· The authors note that 'the national improved latrine coverage in Kenya is estimated at only 30% in 2015'. Could the authors present a more recent estimate? There are similar reports to the referenced reports that were released after 2015.
Response:
We revised from line 71 to line 79 using updated information from World Health Organization; United Nations International Children's Emergency Fund WHO/UNICEF Joint Monitoring Programme for Water Supply, Sanitation, and Hygiene Available online: https://data.unicef.org/topic/water-and-sanitation/sanitation/#data (accessed on 15 August 2023).
· It is reported as 'in Siaya County 73 in western Kenya, which is administratively divided…'. I suggest rephrasing this sentence to two sentences, i.e in Siaya County 73 in western Kenya. Siaya County is administratively divided….'
Response:
We revised according to the comment in line 93 and 95 in the revised version.
· Similarly suggest rephrasing line 74-77 to 'Out of the six sub-counties, two sub-counties (Alego Usonga and Rarieda) were selected from among four sub-counties where the county government had initiated CLTS, and one sub-county (Bondo) was selected from among two sub-counties where CLTS had not been initiated. These sub counties were selected by taking into account the travel distances involved in conducting the survey'
Response:
We revised according to the comment in lines 96 and 100 in the revised version.
· Line 79-80, three villages were randomly selected FROM…
Response:
We revised it according to the comment in line 101 in the revised version.
· Overall, under the study setting, it would be useful for the authors to describe the administrative breakdown, to make it easier for the reader to understand. For example, let the readers know that Counties are divided into sub counties, and villages are the lowest administrative level. This makes it easier for readers to understand the different stages of sampling.
Response:
We added this information in the text in lines 94-95 to make the readers understand the administrative groups in Kenya. With the addition of this sentence, the subsequent explanation that the village is the smallest unit was removed around line 103.
· In addition, please describe the cultural background of the community, for example, before stating that a household of the first wife was interviewed, it would be prudent to explain the cultural setting
Response:
We explained the cultural background in the study area (lines 105-115).
· Line 81-82: What do the authors mean by a household of the first wife? Later on, the authors say 'a compound homestead consists of several households'. Please define a household in your study
Response:
A household is defined as a house with each wife in a polygamous family, and the first wife is defined as the wife who is the head of the household, as determined by the Luo family. This is explained in lines 111 to 115.
· I notice that the document has lots of spelling mistakes, grammatical errors or editorial mistakes. I recommend that the manuscript be checked for grammatical and editorial aspects
Response:
We checked grammatically carefully again.
· Data collection: Please specify how the respondents were sampled/selected
Response:
The households surveyed were the first wife's household of a homestead, and the first wife was selected as the respondent. We added this sentence to line 127-128 additionally.
· Line 106-107, it is not clear what this sentence means. Please clarify whether it means that in the follow up survey the outcome measure was CLTS boosting
Response:
The outcome is to maintain possession of the restroom. The purpose of the follow-up survey is to determine how much toilet retention was maintained by Boost CLTS through follow-up survey, i.e., to determine whether Boost CLTS is effective (lines 139-142).
· Line 119-122; how did you determine the value o livestock? Also, please revise this sentence to bring out the meaning better
Response:
We revised the description to change livestock values to monetary ones in Lines 154- 161 to clarify the calculation. We transferred according to the following exchange rate: 500 KSH per poultry, 4,000 KSH per goat, 3,000 KSH per sheep, 15,000 KSH per cow, 3,000 KSH per pig, and 20,000 KSH per donkey. These monetary values were set up regarding the selling price in the local area. We added an exchange rate of "100 Ksh is equivalent to US$0.7".
· Please revise the term household population to household size
Response:
We replaced them according to the comment (lines 165, 166, and Tables)
· Please describe the type of consent obtained from the respondents
Response:
We obtained verbal consent before we started the interview. We added a sentence below to explain the consent as this: "The data collectors read a written consent form to the participants who gave verbal consent before the start of every interview" in lines 186 -187
· Please describe fully the descriptive analysis done, e.g. no mention of chi square test, yet table 1 shows Chi square results
Response:
We described the descriptive analysis in lines 167-170 in the method section.
Figure 2:
Please correct the grammatical mistakes, e.g Siayaya County
Response:
We revised figure 3 accordingly.
Please specify whether CLTS was implemented at Sub County level or at village level. Previously, it seemed as though it had been implemented at village level
Response:
As we mentioned in lines 136-137, the CLTS was supposed to be done according to village level, therefore, in some sub-county like the Rarieda sub-county. The CLTS situation differed from village to village even though they belonged to the same sub-county.
What do 'households registered' mean? Was there registration of households, or is it in reference to households who were selected for the study?
Response:
Registration means registration and recording on a survey form. In surveying, it means registering for the survey and conducting the survey. The survey targets all households in the selected villages, so no sampling is done. We have revised some descriptions of the household surveyed as lines 197-198.
· Line 181; the authors mention a bivariate model, which was not mentioned in the data analysis. Please clarify
Response:
We are very sorry. This sentence means a "univariate" model for crude Odds ratio calculations. We corrected the term from bivariate to univariate in line 226.
· Table 2 and 3: Include the P-values
Response:
We added the p-values in Tables 2 and 3.
· Line 195: Please stick to the same words. Reading ability or literacy?
Response:
We use "Reading ability" in this paper, so we replace the literacy term with it in the context of the paper.
· Line 206: Check, association between CLTS implementation and what?
Response:
Sorry for the missing term. This sentence indicates an association between CLTS implementation and latrine possession. We corrected the sentence.
· This manuscript only investigates household factors that influence latrine ownership. The authors have clearly spelt out limitations of their study, including the fact that they did not observe the presence of the latrines. I think the title of the study should therefore be about household factors.
Response:
As the reviewer says, we are focusing on collecting and analyzing household information, but we believe that the noteworthy aspect of this study is the repetition of CLTS. Therefore, considering the comments from other reviewers, we would like to change the title to "Effects of Community-Led Total Sanitation (CLTS) Boosting and household factors on latrine ownership in Siaya County, Kenya."
The English needs to be checked for errors.
Response:
We carefully checked the grammatical errors and corrected them.
Reviewer 2 Report
There are aspects that can make the manuscript better, I suggest the authors look at;
Introduction: more details of what CLTS is and the process of implementing it.
Study setting and participants: Line 74-76: the authors states that in Rarieda, the county government had initiated CLTS but in Figure 2 and 3. There is one village in Rarieda (Kametho A) that CLTS had not been initiated.
Actually, when looking at the characteristics of households, there are many factors related to latrine ownership that could be explored through interviews, such as occupation, age, and religion.
Line 97-101: Regarding socioeconomic status (SES), how do you clearly classify a family to be poorest, second, third, fourth?
Results: Figure 2 should be mentioned before Figure 3.
What does “Ref” in Table 2 and Table 3 mean?
Do people in these sub-counties have to pay by themselves in order to build a pit latrine? From the results, although households with CLTS boosting were 7.92 times more likely to own a latrine compared to those with CLTS non-boosting, it seems that SES plays important role as well.
Author Response
We appreciate your valuable comments. We would like to respond to the reviewers' questions and comments and revise the manuscript.
Response to comments from Reviewer 2 There are aspects that can make the manuscript better, I suggest the authors look at; Introduction: more details of what CLTS is and the process of implementing it.
Response:
We appreciate the comments.
We added a CLTS description to the background section (lines 43-55).
Study setting and participants: Line 74-76: the authors states that in Rarieda, the county government had initiated CLTS but in Figure 2 and 3. There is one village in Rarieda (Kametho A) that CLTS had not been initiated.
Response:
Since CLTS was implemented at the village level, one village was selected in the sub-county where CLTS was initiated but not implemented, as the reviewer mentioned. We added this issue in the method section (Lines 102-104).
Actually, when looking at the characteristics of households, there are many factors related to latrine ownership that could be explored through interviews, such as occupation, age, and religion.
Response: We appreciate the comments. While many factors may be involved, as the reviewer mentioned, we did this survey with our design because it is the same tribe, and religiously there is not much diversity.
Line 97-101: Regarding socioeconomic status (SES), how do you clearly classify a family to be poorest, second, third, fourth?
Response: As described in line 162 of the methodology section, the multidimensional data on household possessions, including livestock, obtained from the survey was statistically reorganized into fewer dimensions using principal components analysis (PCA) and divided into quartiles, with the first principal component being the household SES indicator.
Results: Figure 2 should be mentioned before Figure 3.
Response: We appreciate the comment. Figure 2 was first mentioned in Line 102, and Figure 3 was mentioned in Line 199. Therefore, we keep the order in the revised manuscript.
What does "Ref" in Table 2 and Table 3 mean? R
esponse: Ref stands for reference. Reference is the denominator when calculating odds ratios and is the category in which the odds are offered as reference. We have changed to using "Reference" without an abbreviation in the tables.
Do people in these sub-counties have to pay by themselves in order to build a pit latrine? From the results, although households with CLTS boosting were 7.92 times more likely to own a latrine compared to those with CLTS non-boosting, it seems that SES plays important role as well.
Response: Yes. The adjusted odds ratio is calculated by adjusting for SES, i.e., by eliminating the effect of SES. Therefore, the boost effect is still observed independent of SES. Conversely, even after adjusting
for the CLTS boost, SES is still related to latrine ownership, and households with higher SES have higher latrine ownership, suggesting that despite the CLTS effect, households with low SES still need some assistance. This is also noted in the discussion.
Reviewer 3 Report
Please see the attached file.

Author Response
We appreciate your valuable comments. We would like to respond to the reviewers' questions and comments and revise the manuscript.
Response to comments from Reviewer 3
Abstract and Methods
Line 15-16, line 49-65
The authors explained "however, the impact of boosted CLTS on household latrine ownership
has not been adequately evaluated" for the background of this study.
Based on this background, the aim of the study should be to assess the impact of boosted CLTS on household latrine ownership, but the study aim was to investigate "the factors". They should clarify whether the study aim was to assess the impact of the boosted CLTS or to identify factors throughout the manuscript and background, methods, results, and discussion should be linked to each other. Assessing the impact of boosted CLTS on household latrine ownership is one thing, and investigating the factors of latrine possession is another. In this regard, even the title of the manuscript needs to be revised.
Response:
We appreciate the comment.
Although the main focus of this study is to understand the boosting effect of CLTS, it is necessary to incorporate household factors that may affect the effect, i.e., confounding factors, into the model, which is why we conducted this study and report the results.
As for the title of the paper, reviewer 1 commented on the need to change the title of the paper as well. We will change the title as "Effects of Community-Led Total Sanitation (CLTS) boosting and household factors on latrine ownership in Siaya County, Kenya".
In line 50, they described that there is lack of rigorous evidence regarding the impact of CLTS.
Then they continued describing determinants of household latrine, correlation between CLTS implementation and latrine coverage, and also effectiveness of CLTS interventions in
promoting latrine ownership. What particular research gap do they want to highlight and close
by their research? Is it "identifying determinants or factors of household latrine" or "assessing
effectiveness of CLTS"?
Literature review in the introduction is weak and not comprehensive. It is puzzling that they
did not cite some papers of robust methodologies relating to the effectiveness of CLTS
including the trial on CLTS by Pickering in Mali, the trial on CLTS by Abramovsky and the
like. There are many other CLTS or CLTS-like trials investigating its effect.
Response: Thank you for your comment.
We believe that the effectiveness of CLTS has already been just studied and proven as the reviewer mentioned.
On the other hand, we believe that after CLTS, especially in the area of Kenya where we are conducting our study, there are problems with the maintenance of latrines after CLTS and their continued possession afterwards, which is why we have conducted this study to evaluate the boosting effect of CLTS.
Therefore, the purpose of this study is not to examine the health effects of CLTS, itself.
I believe the title of the paper may have caused some confusion. We have changed the title to clarify our aim of the study, and added the paper to the citation we received (line 62).
Methods
If their study aim happens to assess the effects of CLTS, there are many essential components
missing in the manuscript that they did not articulate as follows:
Response:
We appreciate the reviewer's comment.
First, we would like to clarify the purpose of this study, which is to evaluate the impact of CLTS and its promotion on latrine ownership.
With this objective in mind, we would like to address the reviewer's comments and questions.
Think of adding the following sub-headings and detailed descriptions to the methods.
Response:
We prepared this paper according to the journal's template. Therefore, we have decided not to add subheadings, taking into account our response and response to the reviewer.
Study design
Study design must be included in the methods. The authors seemed to try to explain the study
design depicting the Figure 2; however, this figure does not explain the study design and its
rationale to the full extent. For instance, why did they choose some villages that were not
received CLTS intervention? Are they used as a control arm? Some villages did not receive the boosted CLTS intervention. Are they used also as a control group?
Response:
The purpose of selecting villages that did not receive the CLTS intervention was to see the effect of CLTS on latrine ownership using them as the "control group" as the reviewer mentioned in the question above. The same was true for the subsequent boost CLTS intervention; villages that did not receive boost CLTS were used as the control group for the analysis compared to those with boost CTLS>
The results of the analysis showed that CLTS was effective in the baseline survey, and boost CLTS was also effective, but that the CLTS effect waned over the years without boost.
Sampling and sample size
Sampling methods must be elaborated.
Descriptions should be added regarding how they did estimate adequate sample size for this study.
Response:
This paper presents a descriptive epidemiological analysis of data from observational surveys of CLTS intervention status and latrine ownership in the study area and a follow-up survey in the same area. As such, it differs from the research methods used in clinical trials, where the objectives are clearly defined from the outset and conducted for that purpose. Therefore, there was no calculation of sample size, etc. Basically, information was collected from the households of the first wife in all the compounds in the study area.
Exclusion and inclusion criteria should be described.
Response:
As noted above, the inclusion of the first wife's household in all compounds in the study area is only intended to exclude cases of refusal to be interviewed.
Intervention procedures
Detailed explanations on the intervention procedures must be described in methods.
If this study is quasi-experimental study, can they refer to the CONSORT or STROBE guideline?
Please refer to the EQUATOR network website and they could find guidelines. Also, they could refer to the existing literature assessing impact of CLTS or sanitation interventions to check essential components that must be included in methods.
Response:
As stated earlier, this is an observational study, with no intervention; CLTS is already being
implemented by the local government, and we are using the facts of its implementation. Therefore, we see no need to present CONSORT or STROBE guidelines.
If the study aim is to assess the effect of CLTS, they should explain the specific methods to
estimate the effect. The current descriptions might be fit into the aim identifying the factors but not for the effect assessment.
Response:
The main objective of this study was to evaluate the effect of CLTS promotion by comparing the odds ratios of CLTS promotion in villages where CLTS promotion took place with those where CLTS promotion did not take place. We also include the year of first CLTS as a dummy variable as an adjustment factor and find that the effect of CLTS diminishes with the passage of years.
Regarding the figure 1, do the authors have the copyright? Although the programme is publicly available, do they have authenticity? Is it appropriate that the authors describe in a way that they "created the study site map"?
Response:
For Figure 1, we used OpenStreetMap that can be used under the Creative Commons Attribution-ShareAlike 2.0 license. To show the copyright, we put the copyright information in the corner of the map as "(c) OpenStreetMap contributors, CC-BY-SA2.0".
Results
Overall, the contents and structure of the results part depend on the study aim and design.
Currently, based on the current content, the study aim appears to identify factors, not assessing the impact.
Response:
The impact of CLTS is to assess the impact on latrine holding.
Table 1 should explain general characteristics of participants. If they intended to compare the
two groups (for instance, those who received boosted CLTS or not), they should develop a table indicating how balanced the key characteristics of the two groups.
In the table 1, if they indicate the percentage of those who possessed toilet, I don't think they
need to indicate the percentage of those not.
Response:
The purpose of this study was to determine the impact of CLTS on latrine ownership, and Table 1 shows the distribution of factors other than CLTS; the presence or absence of CLTS was not included and has been added in this revision.
Discussion
I would like to hold on reviewing discussion parts since the entire description depend on the
study aim. I hope the authors could revise the discussion parts in alignment with the study aim after they clarify it.
Response:
The purpose of the current study is to evaluate the effectiveness of CLTS on latrine retention. In particular, we want to clarify the effect of boosting CLTS, i.e. that boosting CLTS is necessary for latrine retention. The discussion section is structured accordingly.
Minor comments
There are some minor errors such as missing period [.] between citation number and subsequent sentence (for instance, [1-3] and the following sentence, and [4-7] and the next sentence).
Response:
Appreciate your comments. Correction has been made.
Line 45. I hope they could elaborate collective behavior change or social norm regarding the
CLTS approach. It is not a merely behavior change. In fact, behavior change approach was in
place long before the emergence of CLTS approach.
Response:
Thank you for your important remarks. This study focused on the evaluation of the effects on households. This study focused on the evaluation of the effects on households. The collective behavior changes or social norms regarding is critical of sustainable latrine procession and use. Hence, we added your points in the limitation part. (Line301-303 )
More papers should be cited regarding the study assessing impact of CLTS.
Response: We added some citations according to the comments.
Line 48. Please check whether it isn't over 60 countries.
Response:
Thank you for the suggestion. According to the report of UNICEF, we revised it to "over 60 countries".
Methods
Line 74-78, English should be revised.
Response:
We carefully checked and corrected it.
Legend of the figure 1. The legend should be corrected. (Two "sub-counties" plus 3 additional
"villages" is equivalent to 9 "villages"? I guess the narrative part in the main text is okay but
the legend needs to be revised.)
Legend of the figure 1. The legend should be corrected. (Two "sub-counties" plus 3 additional
"villages" is equivalent to 9 "villages"? I guess the narrative part in the main text is okay but
the legend needs to be revised.)
Response:
We are sorry for that. We revised the legend of figure 1 accordingly.
Line 90. Please attach the questionnaire set.
Response:
We attached the questionnaire sets as supplement documents.
Study design
They have to articulate study design. Is this experimental but not randomized control, right?
The 9 villages are not the same at baseline in terms of the CLTS implementation. How did they reflect this onto the analysis?
Response:
The design used in this study is an observational study of an administrative CLTS intervention, as described in line 93 of the Methods. It is therefore neither an experimental nor a randomised trial.
It is a repeated cross-sectional study that analyses to evaluate the CLTS intervention among villages that were not in the same CLTS situation at baseline, and the subsequent strengthening effect of CLTS.
Furthermore, as CLTS was implemented in each village, we included the village in the analysis as a random effect, as it is likely to be associated with households within the village, and expressed the effect of CLTS as a fixed effect, taking into account the correlation of households within the village.
Was sub-county level clustering effect adjusted?
Response:
No. We adjust for villages as random effects to be adjusted, rather than sub-counties, because CLTS effects and their impacts may spill across villages and there may be correlations between households.
Sub-counties were treated as a single factor, i.e. a fixed effect, to avoid over-adjustment, as the random effect were already accounted for by the villages.
I guess "boosting" is not correct term to the villages where there was no CLTS intervention. If they intend to assess the boosting effect, I think they have to include only those who received CLTS at baseline, and then they have to divide the group into those who received boosted intervention and those who not.
Response:
We appreciate the comment. We defined boosting as the villages that had had first CLTS before baseline survey and received a second CLTS since the baseline survey (mentioned in line 139 - 142). From an analytical point of view, we consider the determination of boosting effects in this study to be valid, as it was carried out as described by the reviewers.
Round 2
Reviewer 2 Report
The background and aim of the study have been clearly defined. The methodology applied is overall correct, the results are reliable and adequately discussed.
The text has to be attentively read before resubmission because there are some typos.
Author Response
Please find our answers in the attached document.
The file name will be automatically assigned as "cover letter" by the system.

Reviewer 3 Report
Please see the attachment.

Author Response
Please find our answers in the attached document.

Round 3
Reviewer 2 Report
I've noticed some issues, such as on line 93, line 305, and line 333. Additionally, some sentences appear informal. I recommend having a native English speaker review the manuscript
Author Response
Response
We appreciate your suggestions.
Correction to line 93 has been done.
We revised the line as “… the impact of CLTS…” in the statement.
Correction to line 305
This statement should read
“Furthermore, and as has been shown in other studies, reading ability…”(Line 304)
Correction to line 333
The statement should read, “Future study is needed to explore the impact of boosted CLTS on collective behavior change or social norms.”(line 332-333)
Also, we corrected several parts of our manuscript according to the suggestion.